# Seroepidemiological Survey of Hepatitis E Virus in Intensive Pig Farming in Vojvodina Province, Serbia

**DOI:** 10.3390/ani15020151

**Published:** 2025-01-09

**Authors:** Diana Lupulović, Marija Gnjatović, Jasna Prodanov-Radulović, Danica Ćujić, Vladimir Gajdov, Milena Samojlović, Tamaš Petrović

**Affiliations:** 1Institute for the Application of Nuclear Energy INEP, Banatska 31b, 11080 Zemun, Serbia; marijad@inep.co.rs (M.G.); danicac@inep.co.rs (D.Ć.); 2Scientific Veterinary Institute “Novi Sad”, Rumenački put 20, 21000 Novi Sad, Serbia; jasna@niv.ns.ac.rs (J.P.-R.); vladimir.g@niv.ns.ac.rs (V.G.); milena.s@niv.ns.ac.rs (M.S.); tomy@niv.ns.ac.rs (T.P.)

**Keywords:** Hepatitis E virus, zoonosis, pigs, serology, ELISA, western blot

## Abstract

Hepatitis E virus (HEV) is widespread in humans and pigs worldwide. HEV-1 and HEV-2 genotypes are restricted only to humans, while HEV-3 and HEV-4 have zoonotic potential and are common in humans and animals. The aim of our study was to determine the presence of anti-HEV antibodies in three hundred naturally infected domestic pigs from three different commercial farrow-to-finish farms in South Bačka on the territory of Vojvodina Province, Serbia. The presence of specific anti-HEV IgG was examined by an in-house ELISA, and doubtful results were retested by western blotting (WB). The overall detected seroprevalence was 40.66%, indicating that the Hepatitis E virus is circulating in the intensive farming pig population in Serbian farms.

## 1. Introduction

Hepatitis E virus (HEV) is the causative agent of Hepatitis E infection, which poses a serious threat to public health. Two epidemiological patterns of the disease exist—endemic and sporadic types. Genotypes 1 (HEV-1) and 2 (HEV-2) infect only humans and are transmitted via the fecal-oral route from contaminated water during large epidemics. The disease is mainly spread in developing countries within Asia and Africa. The mortality rate with HEV-1 and HEV-2 could be between 0.2 and 4%, but, in pregnant women, it can be up to 30% in the third trimester [1,2].

Genotypes 3 (HEV-3) and 4 (HEV-4) have zoonotic potential, with swine being the main reservoir of infection. The first identification of swine HEV was in 1997 in the USA [3]. Since then, pigs have been considered a possible source of infection for human beings. Different strains of HEV have been detected in a wide variety of animal species, such as wild boar, deer, rabbit, camel, etc. [4].

Autochthonous, sporadic cases of HEV-3 have been recorded in many industrialized European countries, where it is increasingly recognized as a foodborne infection through the consumption of undercooked liver, meat, and meat products [5,6,7]. The infection is usually self-limiting, and no specific treatment is necessary. In some cases, the chronicity in immunocompromised patients is reported for the zoonotic HEV-3. Other modes of transmission of HEV-3 and HEV-4 have also been described, including blood transfusion and liver transplantation [8]. Hepatitis E virus can be transmitted via direct contact with swine and represents an occupational hazard. People who are exposed to close contact with pigs, slaughterhouse workers, veterinarians, butchers, and farmers have higher HEV seroprevalence in comparison to the general population [9,10].

Hepatitis E virus is a single-stranded positive-sense RNA particle [11]. According to the newest taxonomy, HEV belongs to the Hepeviridae family, the Orthepevirinae subfamily, and the genus *Paslahepevirus.* Genus *Paslahepevirus* consists of two species, named *Paslahepevirus alci* and *Paslahepevirus balayani* (previously known as Orhoherpesvirus A). Species *Paslahepevirus balayani* encompasses eight genotypes, HEV 1–8 [12]. HEV-1 and HEV-2 are restricted only to humans, while HEV-3 and HEV-4 are common in humans and animals. HEV-5 and HEV-6 were detected in wild boars in Japan. Genotype 7 (HEV-7) was found in a dromedary camel and was linked to chronic viral hepatitis in a patient after liver transplantation [13]. HEV-8 has been discovered in Bactrian camels in China. Although several HEV genotypes have been confirmed so far, only one serotype has been identified [11].

Pigs are the most important reservoir of the hepatitis E virus in nature. They are susceptible to infection but don’t develop clinical signs of the disease. In general, the Hepatitis E virus has little impact on swine health status [3].

Pigs become infected at the age of 2 to 3 months. First, transient viremia occurs, lasting 1 to 2 weeks, and the virus is excreted in the feces for about 3 to 7 weeks [14]. Seroconversion in piglets starts between the 13th and 17th weeks of age when the effect of maternal antibodies ceases, and they develop their own immunity. Anti-HEV IgM and IgA antibodies can be detected first, followed by the appearance of IgG antibodies in the circulation [3]. Throughout their lives, pigs can be reinfected, and seroprevalence increases with age [15].

Determination of HEV seroprevalence in the pig population is of crucial importance for the assessment of epidemiological characteristics on pig farms and, consequently, the risk to human health. The results of seroprevalence vary between countries and farms and depend on the age of tested animals, the number of samples tested, and the methods of analysis. Salines et al. [16] analyzed the seroprevalence on pig farms from 43 European countries and found that the level of anti-HEV antibodies varied between 30 and 98%. In another research, based on 84 studies, the reported HEV seroprevalence on the country level was from 9.90% to 84.02% [17]. In addition, many studies have shown that seropositivity in slaughtering pigs is high and can even reach 100% [18,19]. These data point to a clear risk of an increase in human exposure to hepatitis E infection through contaminated meat.

Pig breeding represents one of the most important branches of agriculture in Serbia. According to data from 2022, 2,667,000 pigs in Serbia are reared yearly. Pig production is mostly concentrated in large farms, with intensive production in Vojvodina Province representing 41.43% of the total livestock production in the country. Also, pork meat production accounted for 58% of the total meat production [20,21]. Several studies were conducted in the last decade on the presence of the Hepatitis E virus in the swine population in Serbia. The established HEV seroprevalence in small backyard farms was 34.6% [22] and 52.25% in wild boar [23]. Furthermore, HEV RNA was identified in 29% of the tested pig livers at a slaughterhouse [24]. In humans, the presence of anti-HEV antibodies was detected in the population of blood donors, with the prevalence rate varying from 15% to 16.9% [25,26]. However, data on HEV infection in pigs in intensive farming in Serbia are scarce. Only one study reported the presence of antibodies against HEV in the restricted area of Belgrade City, with an overall seroprevalence of 55.33% [23].

The aim of our study was to determine the presence of anti-HEV antibodies in three hundred naturally infected domestic pigs from three large farrow-to-finish farms in South Bačka on the territory of Vojvodina Province, Serbia. This cross-sectional study encompasses a comparison of seroprevalence between the different categories of pigs. Our goal was also to characterize the circulation of Hepatitis E infection in the pig population on large farms with intensive farming and to assess the possible threat to human health.

## 2. Materials and Methods

### 2.1. Sampling Strategy and Study Region

In total, three hundred swine blood samples were taken from 3 different farrow-to-finish large farms (A, B, and C, with 1100, 1700, and 700 sows, respectively). One hundred blood samples were collected per farm from 20 animals belonging to 5 different production categories: suckling piglets, weaners, gilts, sows, and fatteners. The selected animals were purebred Landrace and crossbreeds of Landrace and Large White. At the moment of sampling, the animals were apparently healthy, with no evidence of clinical signs of disease. The selected farms were monitored regularly, according to the official veterinary health control programs by the Serbian Veterinary Directorate. All three premises were situated on the territory of Vojvodina Province in the northern part of Serbia (Figure 1).

All pigs included in the survey were properly ear-tagged. Blood sampling was performed by aseptic puncture of the jugular vein. A minimal amount of 10 mL of blood per animal was taken from the pigs in vacutainers without anticoagulant reagents. The samples were delivered to the virology laboratory in a hand fridge with ice, respecting the cold chain of transportation. In the laboratory, the sera were kept at room temperature (20 °C) until the spontaneous coagulation and clot retraction. The separated serum was centrifuged for 10 min at 1500 rpm, collected in microtubes, and then stored at −20 °C until testing.

### 2.2. Serological Examination

The presence of specific IgG antibodies in swine sera against the Hepatitis E virus was examined by in-house ELISA. Doubtful results near the cut-off values were retested by western blotting (WB), which is considered the “gold standard” for confirmation of results.

#### 2.2.1. In-House ELISA

Anti-HEV IgG antibodies have been detected in pig blood sera by a previously described and validated in-house ELISA [27]. Recombinant HEV genotype 3 baculovirus protein Bac1-Δ-ORF2r, obtained by infecting *Trichoplusia ni* insect larvae, was used as an antigen in this test. Briefly, microtiter plates were coated with 15ng/well of antigen, blocked, washed, and incubated with swine sera (diluted 1:10 in blocking solution). Next, an HRP-conjugated goat anti-swine secondary antibody was added to each well. After the addition of the substrate, the reaction was stopped, and the optical density (OD) of the wells was read at a wavelength of 495 nm on an ELISA reader (Asys Expert 96, Biochrom, Cambridge, UK). Positive and negative swine sera pools were used as controls. The cut-off of the assay was established as 2.5 times the OD_495_ value of the negative sera pool. Examined samples with an optical density equal to or greater than the cut-off value was considered positive. In cases where OD values were doubtful, samples were subjected to western blot testing.

#### 2.2.2. Western Blot

Antibodies from blood sera with doubtful in-house ELISA results were analyzed with western blot analysis using the recombinant ORF2 HEV gt3 protein, as described earlier by Jimenez de Oya et al. [27,28].

### 2.3. Statistical Analyses

Statistical analyses and data display were performed with Microsoft Excel 365 and ESRI ArcGIS Pro version 3.03. Pearson’s chi-squared test was used to determine if there was an association between the seroprevalence among pig categories, as well as to compare the seropositivity rate between the farms. A *p*-value set as *p* < 0.05 was considered statistically significant.

### 2.4. Ethical Statement

Blood sampling was conducted in accordance with the recommendation of the Ethical Committee for the Protection of Animal Welfare of the University of Novi Sad. The owners of the farms agreed to have the pigs’ blood sampled for research.

## 3. Results

Table 1 displays the number of HEV-seropositive animals on farms A (37%), B (31%), and C (54%), with the *p*-value indicating the presence or absence of a statistically significant difference in the number of positive samples between the production categories.

The percentage of positivity among the different categories varied significantly, from 0% in suckling piglets on Farm B to 85% in fatteners on Farm C. Seropositivity rate differed significantly between all three examined farms and between categories of suckling piglets, weaners, gilts, and fatteners. Interestingly, a statistically significant difference was not recorded only in the category of sows (*p* = 0.071) (Table 1).

Western blot analysis was performed for the 11 pig sera with doubtful results in the ELISA: six had a positive result, as they reacted with the Bac1-Δ-ORF2r protein, as did the swine positive control serum (See Appendix A).

The mean value of anti-HEV IgG for the pig categories for all three farms is presented in Table 2.

An average of 40.66% (122/300) positivity was recorded for the 3 farms. The seroprevalence rate varied from 33.33% (20/60) in sucking piglets to 61.66% (37/60) in fatteners. The lowest seroprevalence (26.66%,16/60) was detected in the category of sows. The statistical analysis showed a significant difference between all categories of pigs (*p* = 0.00155).

## 4. Discussion

The presence of specific antibodies against HEV was proven by in-house ELISA in all categories of pigs (suckling piglets, weaners, gilts, sows, and fatteners) on tested industrial-type pig farms in the territory of Vojvodina Province, Serbia. The results of our study confirmed that HEV is ubiquitous on examined farms and that the virus circulates intensively in the pig population.

The average rate of anti-HEV antibodies for the three hundred blood samples tested was 40.66%. These data are similar to those previously described in other European countries. For instance, anti-HEV IgG seroprevalence of 48.4% in Spain, 49.8% in Germany, 40% in Bulgaria, 38.14% in France, 43.75% in Lithuania, and 44.06% in Poland has been described on pig farms [17,29,30,31]. However, significantly higher seroprevalences have been reported in China (60%), Greece (80%), and Argentina (80.1%) [32,33,34]. The unequal distribution of HEV seroprevalence can be explained by the different number of analyzed animals, the testing method, sampling plan, farming model, or study design [23,32].

In line with previous studies, our findings also suggest a notable presence of HEV among domestic pigs in Serbia A seroprevalence of 34.6% for anti-HEV antibodies was reported in pigs aged 3 to 4 months from small backyard farms with up to 20 sows [22]. Later on, 55.33% of HEV seropositive animals were detected on farms in the region of Belgrade City [23]. Based on these findings, we can conclude that HEV has been present in a high percentage of domestic pigs in Serbia for at least a decade. However, these data should be considered with caution, as they are limited only to the northern part of the country, while there is no available data for the southern part of Serbia.

A significantly higher seroprevalence was found on Farm C in relation to Farms A and B (Table 1). The HEV positivity rate on Farm C was 54%, while on Farms A and B, it was 37% and 31%, respectively. Farm C has the smallest number of breeding sows (700) compared to farms A and B, with 1100 and 1700 sows, respectively. There are reports that medium-size farms are at greater risk than large-sized farms [15,35,36]. The causes of different percentages of anti-HEV IgG on farms can also be explained by the location of the farm, the different breeding conditions, the number of animals in the facilities, the management on the farm, poor hygiene, and biosecurity measures [19,37]. The HEV seroprevalence in suckling piglets on Farm A was 25% (Table 1), which represents the passive immunity acquired through colostrum intake. This positivity gradually decreased, as only 10% of weaning pigs resulted positive, and increased again to 45% in sows. On Farm B, no seropositive animals were detected among newborn animals. In this case, seroconversion started earlier, which resulted in 25% of HEV-seropositive weaning pigs. Since the presence of HEV IgG on Farm B was found in only 20% of tested sows, we assumed that the passive immunity was weaker and seroconversion in piglets occurred earlier. Our results were in line with other studies. Meng et al. [3] found that piglets from highly seropositive sows had higher antibody levels in the first month of life, which began to decline significantly a few weeks after birth and almost disappeared by 8 to 9 weeks of age. Subsequently, HEV infection and seroconversion occurred at 13 to 14 weeks of age when most piglets developed their own immunity. Piglets originating from the sows with low titers of anti–HEV antibodies were seronegative at birth and seroconversion began earlier, i.e., they were shortly after birth susceptible to HEV infection. De Deus et al. [38] reported that IgG antibodies in piglets originating from highly positive mothers could be detected until the age of 9 weeks, while piglets born from weakly positive mothers had colostral antibodies only in the first 2–3 weeks of life and HEV infection began earlier. At the age of 15 weeks, seroconversion was detected in most animals. Casas et al. [39] described that the duration of maternal immunity in piglets is directly correlated with the level of anti-HEV antibodies in sows, i.e., the higher the level of antibodies in mothers, the longer the duration of maternal immunity.

The results on Farm C for the categories of suckling piglets, weaners, and sows differ from the results on Farms A and B. The presence of antibodies against HEV was detected in 75% of piglets and 85% of weaners, while only 15% of sows and gilts were seropositive. Possible reasons for these results could be the early weaning of piglets, overcrowding of facilities, and mixing of animals that were recorded on this farm, which are considered the main risk factors for infection [19]. In densely populated farms, piglets come into contact with the virus earlier, characteristic of infections transmitted by the enteric route [40,41].

It is interesting to note that HEV seropositivity increased from the category of suckling piglets (33.33%) to the fatteners (61.66%), except for sows, which presented a lower seroprevalence (26.66%) (Table 2). This data can be explained by the fact that gilts and sows are separated into compartments, where the mixing of animals is reduced. Although most authors published data on high seroprevalence in sows [23,39,42,43], there are also results in accordance with our findings. Meng et al. [44] examined many pigs and noted that there was considerable variation among sow categories, Clemente-Casares et al. [45] detected anti-HEV antibodies in 33.3% of primiparous sows and 42.8% of multiparous sows on farms in Spain, and Choi et al. [46] reported that only 8.8% of sows were seropositive on farms in Korea.

In our investigation, the highest level of anti-HEV antibodies (61.66%) was found in fattening pigs, with a range of 45% to 85% between farms. These results are consistent with previously published findings in the literature. For example, Vitral et al. [40] detected 97.3% of seropositive fatteners older than 25 weeks, and Pavia et al. [47] found the highest HEV antibodies rate in pigs older than 4.5 months. Likewise, a Spanish study showed that the proportion of anti-HEV IgG was higher in pigs older than 6 months than in younger ones [48] and Li et al. [17] reported that the seroprevalence rate was 66.20% in 5–8-month-old pigs.

## 5. Conclusions

In conclusion, this study shows that the Hepatitis E virus is widespread on farrow-to-finish farms with intensive production in Vojvodina Province, in the northern part of Serbia. A high circulation of HEV has been detected in all production categories and the overall seroprevalence on the tested farms was 40.66%. These data suggest that pigs are the reservoir of hepatitis E infection in the region and the HEV seroprevalence serves as an indicator of virus circulation within the population.

A higher seroprevalence was found in fatteners than in younger categories of pigs, indicating the possibility that HEV-seropositive animals can enter the slaughterhouses and food chain. Our results confirmed that ELISA is a suitable method for the detection of HEV infection in the pig population, since HEV viremia is short-lived in circulation, which makes diagnosis difficult.

In the future, the mitigation strategy should be focused on more effective biosecurity measures and farming practices to reduce HEV spread within both pig categories and farms. Further analyses should be conducted with the aim of implementing a surveillance program to prevent possible human infection.

## Figures and Tables

**Figure 1 animals-15-00151-f001:**
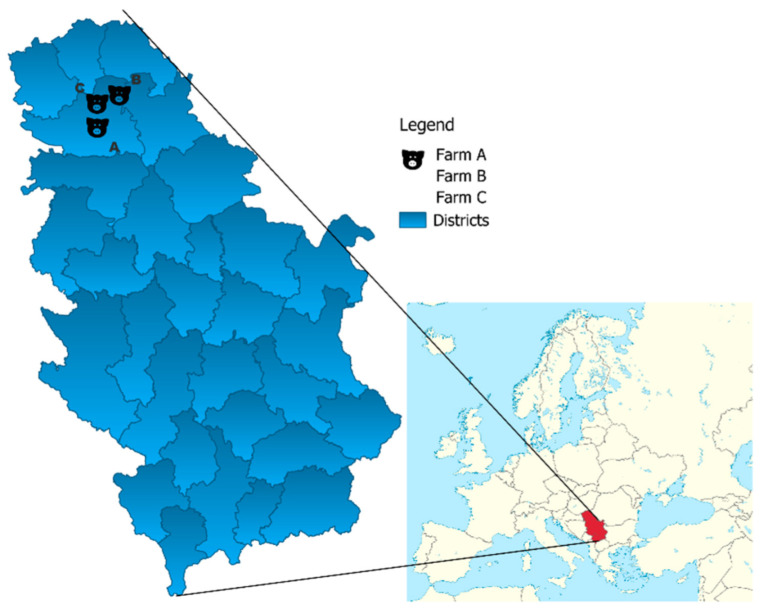
Mapping of locations of swine farms where blood sampling was conducted.

**Table 1 animals-15-00151-t001:** Seroprevalence of anti-HEV IgG on farms A, B, and C by in-house ELISA.

Category	Farm A Pos/Tested (%)	Farm B Pos/Tested (%)	Farm C Pos/Tested (%)	*p*-Value
suckling piglets	5/20 (25%)	0/20 (0%)	15/20 (75%)	0.0000
weaners	2/20 (10%)	5/20 (25%)	16/20 (80%)	0.0000
gilts	12/20 (60%)	11/20 (55%)	3/20 (15%)	0.0070
sows	9/20 (45%)	4/20 (20%)	3/20 (15%)	0.0712
fatteners	9/20 (45%)	11/20 (55%)	17/20 (85%)	0.0255
Total:	37/100 (37%)	31/100 (31%)	54/100 (54%)	0.00274

**Table 2 animals-15-00151-t002:** Seroprevalence of anti-HEV IgG in the different categories of pigs.

Category	HEV IgG (+) Pos/Tested (%)	HEV IgG (−) Neg/Tested (%)	*p*-Value
suckling piglets	20/60 (33.33%)	40/60 (66.67%)	0.00155
weaners	23/60 (38.33%)	37/60 (61.67%)	
gilts	26/60 (43.33%)	34/60 (56.67%)	
sows	16/60 (26.66%)	44/60 (73.34%)	
fatteners	37/60 (61.66%)	23/60 (38.34%)	
Total:	122/300 (40.66%)	178/300 (59.34%)	

## Data Availability

The data presented in this study are available on request from the corresponding author.

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
