# Peer review of "Seroepidemiological Survey of Hepatitis E Virus in Intensive Pig Farming in Vojvodina Province, Serbia"

_animals, 2025, doi:10.3390/ani15020151_

Round 1
Reviewer 1 Report
Comments and Suggestions for Authors
The paper deals with the detection of anti-HEV IgG in sera of pigs housed in 3 farms in the north of Serbia.
Sera were collected from different categories of animals, and the results confirmed a broader circulation of HEV in pigs in Serbia, with an increase in seroprevalence observed in older animals. The discussion is well-written, providing explanations for the results obtained. However, the paper would have a greater impact on topic if the presence of the virus had been investigated. Why did the authors not investigate at least the presence of IgM? The same ELISA could be used with by replacing the secondary antibodies. This information, which is less commonly available for HEV, would represent an important finding as it indicates recent infection.
The paper only needs minor revisions. Please find below some suggestions
line 24: please add a sentence that clearly explain that two epidemiological patterns of the disease exist, specifying that the zoonotic virus causes sporadic cases and small outbreaks in high income countries. The same aspect should be better explained at line 42. I’d suggest split the whole paragraph so to clearly explain that the mortality rate reported and the risk in pregnant women is only referred to genotype 1 and 2 as well as the chronicity in immunocompromised patients is reported for the zoonotic genotype 3
line 45: remove “isolation”
line 64: please specify “host of HEV-3 and HEV-4”
line 135: can you please include more details about the ELISA test? At which dilution were the sera tested? Did you also include positive and negative control?
line 144: use the term “antibodies” , it is better than “proteins”
line 169: better specify that the differences among sow are not statistical support (p 0.071)
line 207-208: is there any difference in the number of pigs housed in the three farms? do you exclude that the higher seroprevalence in the farms C is linked to the number of animals housed? at line 108 you could add the information on total number housed in the farms
line 264: the detection of IgG antibodies indicates prior exposure of the animals to the virus, but it does not necessarily signify an active stage of infection. Consequently, the risk to humans is not directly proportional to the seroprevalence. However, I agree that the seroprevalence does serve as an indicator of virus circulation within the population. Please revise the sentence specifying better your conclusion
Author Response
Dear Reviewer 1,
We have made changes to the manuscript in accordance with your comments. You will find our responses in a separate Word file attached.
Best regards,
Diana Lupulović
Corresponding author

Reviewer 2 Report
Comments and Suggestions for Authors
The communication by Lupulović et al. has investigated the seroprevalence of Hepatitis E virus (HEV) in intensive pig farming in Vojvodina Province, Serbia. The study concludes that HEV is widespread in intensive pig farming in the study region and recommends implementing biosecurity measures and surveillance programs to prevent zoonotic transmission. While without specific HEV genotype analysis and HEV seroprevalence study in farm workers, it is hard to define its impact on public health. Continued surveillance and seroprevalence studies of HEV in humans, especially those in close contact with pigs (e.g., swine farm families and workers and slaughterhouse workers), may provide a clearer answer in the future.
Specific comments:
1. What are the criteria for farm selection? Were the farms randomly selected, or were they chosen based on convenience or other factors? Knowing the selection process would improve confidence in the results.
2. As you mentioned in lines 13-14, “HEV-3 and HEV-4 have zoonotic potential and are common for humans and animals.” While in this study, specificity of the HEV genotype(s) was not involved. Knowing the genotypes present would greatly enhance the risk assessment for human infection.
3. Did you do any HEV seroprevalence study in farm workers to mitigate zoonotic transmission risk to humans?
4. Where are the actual data of in-house ELISA and WB? Suggest adding actual data as supplementary.
Author Response
Dear Reviewer 2,
We have made changes to the manuscript in accordance with your comments. You will find our responses in a separate Word file attached.
Best regards,
Diana Lupulović
Corresponding author

Round 2
Reviewer 2 Report
Comments and Suggestions for Authors
No further comments.